# A Holistic Wetland Ecological Water Replenishment Scheme with Consideration of Seasonal Effect

**Haiyan Duan [1], Menghong Xu [1], Yu Cai [1], Xianen Wang [1], Jialong Zhou [1] and Qiong Zhang [2],***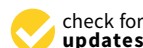

[1]  Key Laboratory of Groundwater Resources and Environment of Ministry of Education, College of Environment and Resources, Jilin University, 2699 Qianjin St., Changchun 130012, China; duanhy1980@jlu.edu.cn (H.D.); xumh16@mails.jlu.edu.cn (M.X.); petrel1980@126.com (Y.C.); wxen@jlu.edu.cn (X.W.); zhoujialongjlu@126.com (J.Z.)

[2]  Civil and Environmental Engineering Department, University of South Florida, 4202 E. Fowler Ave., Tampa, FL 33620, USA

\*  Correspondence: qiongzhang@usf.edu

**Abstract:** Wetland ecological water replenishment becomes necessary in most developing countries. A holistic water replenishment scheme considering both wetland ecosystem services and irrigation requirement is needed for river water reallocation. A framework was developed in this study to calculate wetland ecological water demand (WD), river water supply capacity (RSC) and the benefit of wetland ecological water replenishment and crop irrigation with consideration of the seasonal effects. The Xianghai wetland and the Taoerhe irrigation district (TID) were considered as the study area to investigate various wetland ecological water replenishment schemes (WRS). The results showed that the WRS, considering both wetland function and agricultural irrigation, has the highest overall benefit compared to the schemes with a single focus (either wetland or irrigation). In addition, the WRS design must consider the seasonal effect because of seasonal variation of rainfall, crop growth, and wetland plants and animals' growth. The WRS design with consideration of seasonal effect not only increased the total value of river basin from \$74.83 million to \$104.02 million but also balanced the benefit between TID and wetland while meeting wetland WD. This study offers a decision-making framework of developing a holistic WRS considering benefits from multiple water users and seasonal variation.

**Keywords:** wetland; ecological services; seasonal variation; irrigation; water allocation

## 1. Introduction

Wetlands are irreplaceable and important ecosystems with many hydrologic, biological, and ecological functions [1–3]. However, due to natural factors such as climate change and the human activities such as industry and agriculture water uses, water flow into wetlands is consistently decreasing [4]. As a result, wetland area has been substantially shrinking, especially in China, which has reduced by 33% from 1978 to 2008 [5]. Ecological water replenishment becomes necessary to maintain wetland existence and function [6–8].

Water demand (WD) for wetland restoration put additional stress to already scarce water resources, especially in developing countries. The river waters are supplied to meet the demand from other users such as agriculture, industry, and municipalities at the same time. If the river water is reallocated to restore wetlands, the wetland ecological water replenishment scheme (WRS) should consider not only wetland ecological water demand (WD), but also the river water supply capacity (RSC) and the affected water users. In recent years, an increasing number of ecological and environmental studies have been conducted on the environmental flow requirements for freshwater

wetlands [9–14]. Most of these studies focused on the methods of determining WD such as the wetted-perimeters method [15], the wetland classification calculation method [16], the remote sensing and GIS method [17,18], the ecological hydrological analysis method [19] and the wetland ecological function method [20]. However, these studies considered only water requirements for wetland without taking into account of the constraints of water resources and competition from other users. Although some of the studies focused on the cost-benefit analysis of diverting water for agricultural use to wetlands [21], the benefits to both agriculture and wetland were not evaluated. In addition, the RSC and the seasonal effects were not considered in those studies. Such factors will impact the river water reallocation especially in the regions where there is significant competition for scarce water resources.

Wetlands have different WDs in the different seasons because of the growth of plants and animals. On the other hand, rivers have different water supply capacity in the rainy season and dry season. The quality of water available for wetland replenishment also varies with seasons because of the non-point source pollution caused by stormwater runoff, and the water quantity and quality are naturally coupled in different seasons. In addition, WD from other users, such as agriculture, has a significant seasonal variation. Since approximately 60–70% of all the freshwater worldwide is used for agricultural irrigation [22–24], the affected area due to water diversion for wetland restoration is mainly farmland. Since the growth of the crops is highly dependent on temperature and water availability, the impact of water shortage on crop yield and crop quality will vary in different seasons [25–28]. Such seasonal variations in water supply capacity, wetland WD, and crop yield and quality have not been incorporated in the design of the WRS to achieve a win-win situation for wetland and irrigation management.

Hence, this research aims to develop a holistic WRS with consideration of both wetland ecological services and irrigation requirement under the seasonal variations of water supply, demand, and the associated benefit of the water reallocation. This research offers a decision-making framework of water resource allocation in the basin where there is a conflict between wetland water use and agricultural irrigation demand. This research is intended to inform researchers, resource managers, and policy makers on the water replenishment strategies for wetland restoration.

## 2. Methodology

The analysis of the WD and the benefits to both the wetland and the irrigation area followed the steps shown in Figure 1. The wetland ecological WD and the water requirement for wetland replenishment were calculated first. The RSC in different seasons was determined with the consideration of natural inflow and environmental flow requirement. The difference between the water supply capacity and the water needed for agricultural irrigation was defined as the surplus water capacity. In the case that the surplus water capacity was more than wetland replenishment requirement, the surplus river water was directly delivered to wetland. Otherwise, the river water used for irrigation was diverted to replenish wetland and the ground water was used for irrigation instead of river water. Therefore, the water allocation led to the different benefits to wetland and farmland.

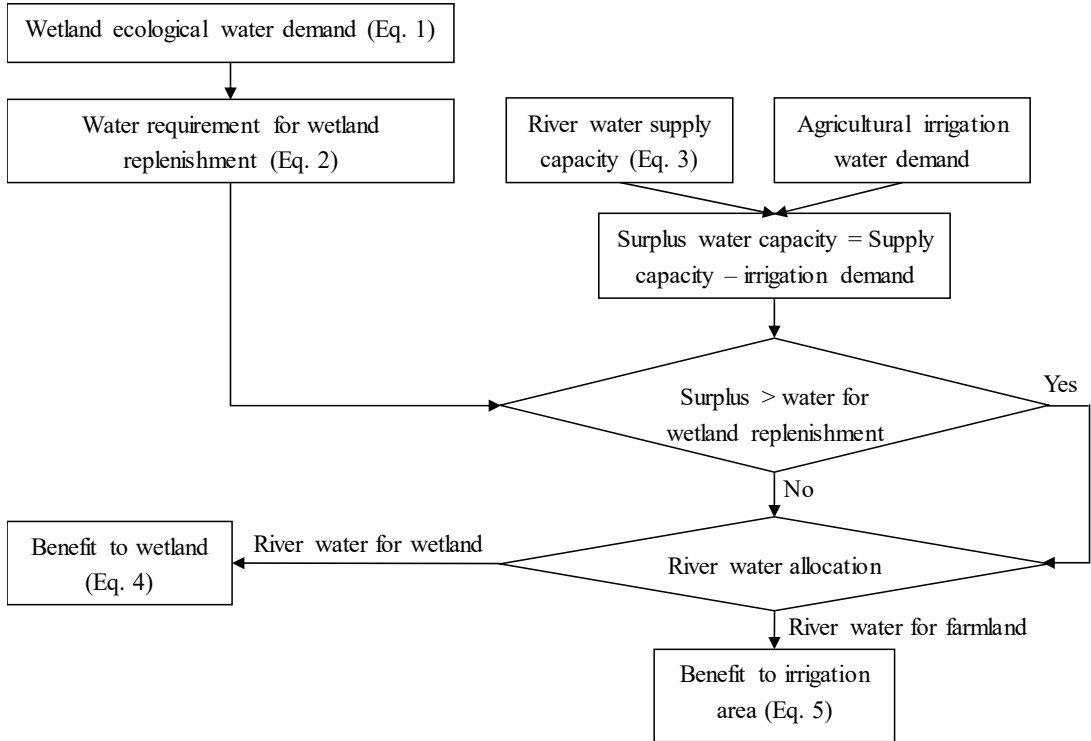

**Figure 1.** Flowchart for the calculation steps involved in the study.

## 2.1. The WD and Water Replenishment

The WD refers to the amount of water that wetlands need for the development and to protect biodiversity [29]. The critical thresholds have been used for the WD which include the maximum wetland ecological water demand (Max-WD), the baseline wetland ecological water demand (B-WD) and the minimum wetland ecological water demand (Min-WD) [16]. The Max-WD is the amount of water required to maintain the wetland ecosystem in the best state. The B-WD is the amount of water required to ensure that the ecological environment does not deteriorate further. The Min-WD is the minimum amount of water for the wetland to maintain its own development, below which the wetland will be degenerated or even disappeared. The Max-WD and the Min-WD provides the upper bound and the lower bound of wetland water demand.

The WD mainly consists of two parts, which are the water demands of wild habitats and plants [29]. Wild habitat WD refers to the water required for habitat and reproduction of fishes and birds. The WD can be calculated using Equation (1) [29]:

$$Q_{i,k} = (E_i \cdot \alpha \cdot \gamma - R_i) \cdot A_k \cdot 10^{-3} \tag{1}$$

where, $Q_{i,k}$ is the WD (km$^3$), $k$ represents the different thresholds (1 for the Max-WD, 2 for the B-WD and 3 for the Min-WD); $i$ is the corresponding time (month or season); $E_i$ is the evaporation (mm) in the season $i$; $\alpha$ is the coefficient of water evaporation (dimensionless); $\gamma$ is the coefficient of plant evapotranspiration (dimensionless); $R_i$ is the long-term average rainfall (mm) in the month or season $i$; $A_k$ is the area of lakes, reservoirs and plants of a wetland (km$^2$) ($k$ = 1, 2, and 3 representing the largest area, average area, and the core area of the wetland, respectively).

The water replenishment ($Q_{R,i,k}$) is the amount of water diverted from the water sources to a wetland for its maintenance and can be calculated using Equation (2). Equation (2) is derived based on flow balance with consideration of the water loss.

$$Q_{R,i,k} = (Q_{i,k} - Q_{NW,i})/(1 - \xi) \tag{2}$$

where, $Q_{NW,i}$ is the natural inflow to the wetland in the month or season $i$ (km$^3$); $\xi$ is the water loss coefficient because of the leakage and evaporation along the way.

*2.2. The RSC in Different Seasons*

In this study, another main water user considered is agriculture since 60–70% of freshwater in a river is used for agricultural irrigation [23]. As a result, the RSC accounts for the amount of water that can be allocated for wetland and agricultural use. The RSC is calculated using Equation (3):

$$Q_{r,i} = Q_i - Q_{eco,i} - Q_{NW,i} \tag{3}$$

where $Q_{r,i}$ is the RSC in the month or season $i$ (km$^3$); $Q_i$ is the total amount of water in the river in the month or season $i$ (km$^3$); $Q_{eco,i}$ is the amount of water required to maintain environmental flow of the river in the month or season $i$ (km$^3$).

*2.3. The Benefit of River Water Diversion*

2.3.1. The Benefit to Wetland

Water replenishment can protect and recover the wetland ecosystem to promote the ecosystem services. In this study, the value of the ecosystem services is used to quantify the benefit of water replenishment of wetland. Since the purpose of this study is to design a holistic WRS rather than an accurate estimate of the wetland ecosystem service value, the "*wetland ecosystem functional value coefficient method*" [30] was used to estimate the value of the ecosystem services. This method calculates the functional value of a wetland based on the acreage of the wetland and the value coefficient. The value coefficient refers to the ecosystem service value provided by the unit area of the ecosystem [2,31,32]. A wetland can provide various ecosystem services, such as climate regulation, water storage soil formation, and conservation, waste treatment, food production, raw materials, recreation and culture, and so on. The value coefficient is specific for the service provided and varies with climate, geographical location, biological structures, and biomass. The benefit to wetland ecosystem services ($B_W$) is calculated using Equation (4) [30]:

$$B_W = \sum_{g=1}^{5} \delta_g \cdot \Delta W_A \tag{4}$$

where $\delta_g$ is the ecosystem service value coefficient ($/km$^2$), $g$ represents the type of ecosystem services; $\Delta W_A$ is the acreage difference of wetland before and after water replenishment (km$^2$).

2.3.2. The Benefit to Irrigation Area

To evaluate the benefit to the farmland, three criteria were considered in this study including crop output, crop quality, and the increased water cost.

Since the source of water has different effects on crop output at different crop growth stage [33], the benefit of using river water for irrigation varies monthly. For example, the crop yield using groundwater for irrigation during the growth stages is typically lower in the study area due to the low water temperature and high alkalinity and hardness. In this study, the *Jensen model* [34] was used to analyze the impact of the source of water on crop output, which is a multiplication model considering both the direct effect of the source of water on crop outputs at different stages and the indirect effects on the final yield of a crop.

In addition, irrigation using groundwater in the study area has a negative impact on crop quality [35]. According to the "*Barrel Theory*" [36], a crop has the desired grain quality only when it is irrigated with the appropriate source of water without water shortage at any growth stage; otherwise, the quality will be compromised regardless of the growth stage at which water shortage occurs. In the study area, when surface water is not sufficient for irrigation, groundwater will be used to meet the irrigation WD. The impact of using groundwater on crop quality was evaluated using the surface water deficiency ratio (DR) [7] in affected farmland in this study. The water DR is the ratio of water deficit (the difference between irrigation WD and surface water supply) to total irrigation WD, which represents the degree of surface water shortage in irrigation.

If the river water is diverted to wetlands, some farmlands must use groundwater instead of surface water for irrigation, which introduces additional costs of implementing groundwater irrigation system such as drilling a well, installing a pump, and associated energy consumption.

Considering all three aspects, the benefit to the farmland ($B_{IA}$) is formulated as Equation (5):

$$B_{IA} = \sum_{j=1}^{2} \varphi_j \cdot S_A + I_A \cdot \Delta c \tag{5}$$

where

$$\varphi_1 = \beta \times \Delta Y_{s-g} \times P_s$$

$$\varphi_2 = \lambda \times Y_s \times \Delta P_{s-g}$$

$$\beta = 1 - \prod_i \sigma_i^{\theta_i}$$

$$\lambda = 1 - Max(\sigma_i)$$

$\varphi_j$ is the adjustment coefficient ($\$/km^2$) for crop output (j = 1) and crop quality (j = 2) in month *i* (the growing period); $S_A$ is the acreage of affected farmland ($km^2$); $I_A$ is the acreage of irrigated area ($km^2$); $\Delta c$ is the cost difference between surface water irrigation and groundwater irrigation ($\$/km^2$); $\beta$ is the influence coefficient of crop output, which stands for the affected degree on the crop output in different seasons; $\Delta Y_{s-g}$ is the crop output difference between surface water irrigation and groundwater irrigation ($kg/km^2$); $P_s$ is the crop price under the conditions of surface water irrigation ($\$/kg$); $\lambda$ is the influence coefficient of crop quality, which stands for the affected degree on the crop quality in different seasons; $Y_s$ is the theoretical crop output under the conditions of surface water irrigation ($kg/km^2$); $\Delta P_{s-g}$ is the crop price difference between surface water irrigation and groundwater irrigation ($\$/kg$); $\sigma_i$ is DR in the affected farmland (%); $\theta_i$ is the water sensitivity index of crop in the state of water shortage in month *i*.

*2.4. Scenario Description*

Different WRS have different impacts on the wetland and the farmland. Four different scenarios are considered in this study:

(I) Priority to meet irrigation need

In this scenario, it is assumed that the irrigation water should be met first without considering the WD. The rest of river water can be introduced to the wetland. This scenario is usually happening in most of the agriculture-based developing countries.

(II) Priority to meet wetland water need

In this scenario, the wetland ecological WD should be met first, and the farmland receives the rest of river water. Generally, this type of replenishment takes place only when the wetlands are facing the danger of ecosystem damage. In some developing countries, this is called the emergency replenishment.

(III) Meet the baseline ecological WD without consideration of seasonal effect

As defined early, the B-WD is the amount of water required to ensure that the wetland will not deteriorate further. In this scenario it is assumed that the total B-WD will be met by diverting the river water equally in the months of the year to the wetland without consideration of the seasonal impacts on crops. If the river water capacity in a month is less than the designed monthly water diversion, the river water should be all introduced to the wetland in that month. The remaining required water will be evenly introduced in other months during which the river supply capacities are higher than the water diversion. This scenario just considers the WD but not seasonal effect on wetland ecosystem and agricultural crops.

(IV) Meet the appropriate ecological WD with consideration of seasonal effect

In this scenario, it is assumed that the total B-WD should be met; however, the WRS was determined according to the monthly surface water DR of the irrigation area. In this case, the monthly Min-WD must be met first for protecting the core area of wetland. In addition, then, the river water will be introduced to the wetland to meet the total B-WD with consideration of the seasonal impacts of water diversion on crops. Since the surface water DRs in the irrigation area vary monthly, the water will be diverted to the wetland either in the month of low water DR or in the month with less impact on crops. When the water DRs are the same over months, the water can be distributed to wetland evenly by month. This scenario not only considers the WD of wetland but also consider the seasonal effect on wetland ecosystem and agricultural crops.

## 3. Study Area

*3.1. Xianghai Wetland*

The Xianghai wetland is in Jilin Province, northeast China, at 122°05′–122°35′E, 44°50′–145°19′N (Figure 2). There are three rivers around the Xianghai wetland which are Huolin River, Emutai River, and Taoerhe River. The Xianghai wetland is the flood discharge area of the Taoerhe River and the irrigation area named the TID is in the downstream of the Taoerhe River. The water from the Taoerhe River is mainly used for agricultural irrigation.

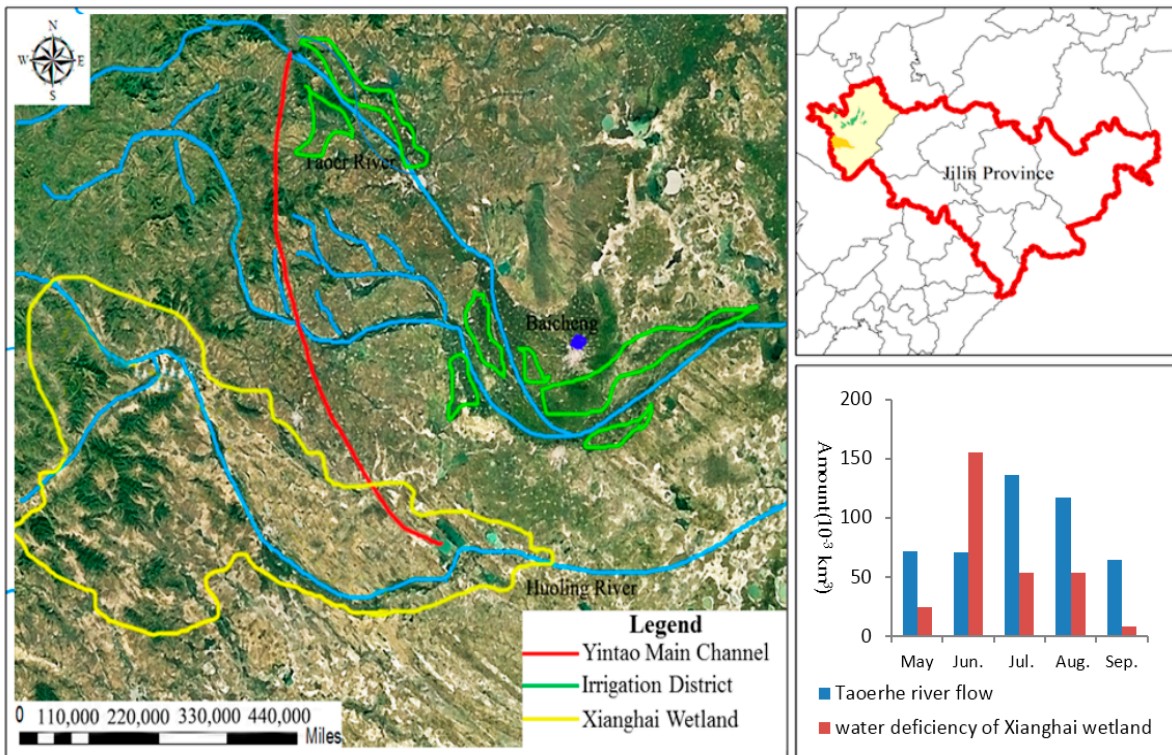

**Figure 2.** The overview of the study area with the Xianghai wetland (the largest area) and the Taoerhe irrigation district.

The total acreage of the Xianghai wetland is $1.055 \times 10^5$ km$^2$. The acreage of the core area in the Xianghai wetland is $2.899 \times 10^4$ km$^2$. The Xianghai wetland is listed as one of the *Internationally Important A-Class Nature Reserves* by the *Worldwide Fund for Nature (WWF)*, with abundant natural resources including 537 species of plants and 292 species of animals.

The Xianghai wetland belongs to a semi-arid climate. From October to April, the average temperature is 10 degrees below zero. According to the "*Baicheng City Water Resources Public Report of China*", a government document. In the dry years (the guaranteed rates of 75%) since 1980, the annual average precipitation is 360.6 mm and the annual average evaporation is 1686.5 mm. There is a reservoir named Xianghai Reservoir in the wetland which can store water. In recent years, influenced by global climate change, the water conservation project of the upstream Huolin River and the water consumption for agricultural irrigation, the average natural inflow into the Xianghai wetland has been decreasing, leading to a sharp decline of the wetland area. In a typical dry year (the guaranteed rates of 75%), the natural inflow to the Xianghai wetland is 0.104 km$^3$. In the study area, the frozen period (the months of January, February, March, April, November, and December) requires no irrigation and the crops are harvested in the month of October. Therefore, the water diversion for irrigation was considered only for the period of May to September in this study.

### 3.2. River Water Supply of the Xianghai Wetland

The Taoerhe River flow from north to south and is the main water source of the Xianghai wetland in a dry season. As shown in Figure 1, the Huolin River flow from east to west through the south section of the Xianghai wetland. In the north, the Taoerhe River discharges its floodwater into the wetland through the main canal of *Yintao*. The canal is also used by the Emutai River for discharging floodwater during the rainy season. However, in a year of drought, the flow of Huolin River disappears when it arrives in the Xianghai wetland. The Emutai River is a seasonal stream, which suffers from water shortage throughout the year. Compared with the Huolin River and the Emutai River, the

water of the Taoerhe River is relatively abundant, which can be considered as the replenishing water source for the Xianghai wetland. The water can be diverted to the Xianghai wetland through the main canal of *Yintao* with a water loss coefficient of 56.81%. As mentioned above, the water allocation from the Taoerhe River to the wetland occurs during the non-freezing period from May to September. The river water supplies during May to September are $7.178 \times 10^{-2}$ km$^3$, $7.128 \times 10^{-2}$ km$^3$, 0.133 km$^3$, 0.117 km$^3$ and $6.48 \times 10^{-2}$ km$^3$, respectively.

### 3.3. Taoerhe Irrigation District (TID)

The TID lies in the downstream of the Taoerhe River, which is a large-scale irrigation district of paddy in China, built in 1993 with an area of $3.32 \times 10^5$ km$^2$. The main crop in the district is rice. Since 1998, the water of the Taoerhe River is mainly used for the agricultural irrigation in the TID during the non-flooding period. Water is diverted from the Taoerhe River to the TID in the months of May to September every year with a water loss coefficient of 50%. According to the report of "*Irrigation Water Planning*" from the Taoerhe irrigation district Bureau, the annual irrigation water requirement is 0.153 km$^3$ with significant seasonal variation ($1.725 \times 10^{-2}$, $7.776 \times 10^{-2}$, $2.678 \times 10^{-2}$, $2.678 \times 10^{-2}$, and $4.32 \times 10^{-3}$ km$^3$ from May to September). Considering the water loss, the actual amount of water diverted from the Taoerhe River is doubled ($2.45 \times 10^{-2}$, 0.156, $5.356 \times 10^{-2}$, $5.356 \times 10^{-2}$, and $8.64 \times 10^{-3}$ km$^3$ from May to September). After water allocation, if the maximum wetland ecological water demand (Max-WD) is met, Xianghai wetland will have the largest area of $1.055 \times 10^5$ km$^2$. If the baseline wetland ecological water demand (B-WD) is met, Xianghai wetland will have the average area of $6.423 \times 10^4$ km$^2$. If the minimum wetland ecological water demand (Min-WD) is met, Xianghai wetland will only have the core area of $2.899 \times 10^4$ km$^2$.

### 3.4. Data Sources

The data and parameters required in Equations (1)–(5) are compiled in Table 1. The data on the Xianghai wetland (e.g., acreage) and water flows were obtained from the government report and the monitoring data. Some data, such as ecosystem service value coefficients, were calculated based on the values from the literature and unit conversion (e.g., from RMB to dollars for ecosystem service value coefficients). Some data such as the benefit of using surface water in irrigation area as a long-time average data were directly provided from *TID Authority*.

**Table 1.** Data and parameters used in the study.

| *Data or Parameters* | | *Values* | *Units* | *References* |
|---|---|---|---|---|
| *Acreage of the Xianghai wetland* ($A_k$) | | | | |
| *The largest area* (k = 1) | Lakes acreage | $4.427 \times 10^3$ | $km^2$ | The report of "Remote Sensing Dynamic Analysis Report of Land Use in Songnen Plain Wetland Nature Reserve" which come from China Songliao River Basin Commission. |
| | Reservoirs acreage | $2.376 \times 10^4$ | | |
| | Plants acreage | $2.403 \times 10^4$ | | |
| | Others | $5.325 \times 10^4$ | | |
| *The average area* (k = 2) | Lakes acreage | $1.218 \times 10^3$ | $km^2$ | |
| | Reservoirs acreage | $6.538 \times 10^3$ | | |
| | Plants acreage | $2.403 \times 10^4$ | | |
| | Others | $3.244 \times 10^4$ | | |
| *The core area* (k = 3) | Acreage of lakes and reservoirs | $1.517 \times 10^3$ | $km^2$ | |
| | Plants acreage | $1.084 \times 10^4$ | | |
| | Others | $1.663 \times 10^4$ | | |
| *Environmental flow* ($Q_{eco,i}$) | | $2 \times 10^{-3}$ | $km^3$ | [7] |
| *Coefficient* | Water evaporation ($\alpha$) | 0.5–0.6 | dimensionless | [37] [38] |
| | Plant evapotranspiration ($\gamma$) | 1.1–2.5 | | |
| *Natural water in the dry year* ($Q_{NW,i}$) *(guaranteed rates of 75%)* | May | 0 | $km^3$ | Monitoring and statistical data provided by Hydrology Bureau of Baicheng City |
| | June | 0 | | |
| | July | $2.3 \times 10^{-2}$ | | |
| | August | $3.8 \times 10^{-2}$ | | |
| | September | $4 \times 10^{-2}$ | | |
| *The benefit of using surface water in irrigation area* | The theoretical rice output under the conditions of surface water irrigation ($Ys$) | 8000 | kg | Provided by Taoerhe Irrigation District Authority |
| | The rice output difference between surface water irrigation and groundwater irrigation($\Delta Ys\text{-}g$) | $2.2 \times 10^5$ | $kg/km^2$ | |
| | The c rice price under the conditions of surface water irrigation ($Ps$) | 0.55 | $/kg | |
| | The rice price difference between surface water irrigation and groundwater irrigation($\Delta Ps\text{-}g$) | 0.09 | $/kg | |
| | The cost difference between surface water irrigation and groundwater irrigation($\Delta c$) | $9.848 \times 10^3$ | $/km^2 | |
| *Ecosystem service value coefficients* ($\delta g$) | Climate regulation | $1.613 \times 10^4$ | Million $/km^2 | [30,39,40] |
| | Water storage | $9.046 \times 10^4$ | | |
| | Soil formation and conservation | $5.033 \times 10^3$ | | |
| | Biodiversity maintenance | $4.061 \times 10^4$ | | |
| | Recreation and culture | $2.556 \times 10^4$ | | |
| *The water sensitivity index of rice* ($\theta i$) | Mid-tilling (MT) stage | 0.2826 | dimensionless | [33] |
| | Elongation stage | 0.6284 | | |
| | Efflorescence and filling stage | 0.4406 | | |
| | Maturation stage | 0.1086 | | |

## 4. Result and Discussion

### 4.1. Xianghai Wetland Water Demand and the Taoerhe River Water Supply Capacity

The results of the monthly Xianghai WD are presented in Figure 2a. The total Max-WD, B-WD, and Min-WD were respectively 0.3144 km$^3$, 0.191 km$^3$ and 9.064 × 10$^{-2}$ km$^3$. Considering the natural flow (0.104 km$^3$) and the water loss rate (56.81%), the Xianghai wetland needed to introduce 0.486 km$^3$ and 0.202 km$^3$ water from the Taoerhe River to meet the Max-WD and the B-WD in a typical dry year.

After deducting environmental flow and natural inflow, the Taoerhe RSC was 0.347 km$^3$ over the months of May to September and the monthly RSC was successively 6.978 × 10$^{-2}$km$^3$, 6.928 × 10$^{-2}$ km$^3$, 0.108 km$^3$, 7.705 × 10$^{-2}$ km$^3$ and 2.28 × 10$^{-2}$ km$^3$ (Table 2). The Taoerhe RSC has been unable to meet the WD of the TID (0.306 km$^3$) and the Xianghai wetland (0.486 or 0.202 km$^3$) simultaneously.

**Table 2.** The ecological water replenishment Scheme (km$^3$).

| | | May | June | July | Aug. | Sep. | Total |
|---|---|---|---|---|---|---|---|
| *River water supply capacity of the Taoerhe River* | | 6.978 × 10$^{-2}$ | 6.928 × 10$^{-2}$ | 0.108 | 7.705 × 10$^{-2}$ | 2.28 × 10$^{-2}$ | 0.347 |
| *Water requirement from the Taoerhe irrigation district* | | 3.450 × 10$^{-2}$ | 0.156 | 5.356 × 10$^{-2}$ | 5.356 × 10$^{-2}$ | 8.64 × 10$^{-3}$ | 0.306 |
| *Xianghai wetland water requirement for the minimum wetland ecological water demand* | | 4.971 × 10$^{-2}$ | 3.989 × 10$^{-2}$ | 0 | 0 | 0 | — |
| *Scenario I* | Taoerhe Irrigation District | 3.45 × 10$^{-2}$ | 6.928 × 10$^{-2}$ | 5.356 × 10$^{-2}$ | 5.356 × 10$^{-2}$ | 8.64 × 10$^{-3}$ | 0.220 |
| | Xianghai wetland | 3.528 × 10$^{-2}$ | 0 | 5.429 × 10$^{-2}$ | 2.349 × 10$^{-2}$ | 1.416 × 10$^{-2}$ | 0.127 |
| | Water deficiency ratio | 0.00% | 0.00% | 0.00% | 0.00% | 0.00% | |
| *Scenario II* | Xianghai wetland | 6.978 × 10$^{-2}$ | 6.928 × 10$^{-2}$ | 0.108 | 7.705 × 10$^{-2}$ | 2.28 × 10$^{-2}$ | 0.347 |
| | Taoerhe Irrigation District | 0 | 0 | 0 | 0 | 0 | 0 |
| | Water deficiency ratio | 100.00% | 44.55% | 100.00% | 100.00% | 100.00% | — |
| *Scenario III* | Xianghai wetland | 4.476 × 10$^{-2}$ | 4.476 × 10$^{-2}$ | 4.476 × 10$^{-2}$ | 4.476 × 10$^{-2}$ | 2.28 × 10$^{-2}$ | 0.202 |
| | Taoerhe Irrigation District | 2.502 × 10$^{-2}$ | 2.452 × 10$^{-2}$ | 6.309 × 10$^{-2}$ | 3.229 × 10$^{-2}$ | 0 | 0.145 |
| | Water deficiency ratio | 27.49% | 28.78% | 0.00% | 39.72% | 100.00% | — |
| *Scenario IV* | Xianghai wetland | 4.971 × 10$^{-2}$ | 3.989 × 10$^{-2}$ | 6.369 × 10$^{-2}$ | 3.289 × 10$^{-2}$ | 1.568 × 10$^{-2}$ | 0.202 |
| | Taoerhe Irrigation District | 2.007 × 10$^{-2}$ | 2.939 × 10$^{-2}$ | 4.416 × 10$^{-2}$ | 4.416 × 10$^{-2}$ | 7.12 × 10$^{-3}$ | 0.145 |
| | Water deficiency ratio | 41.83% | 25.65% | 17.54% | 17.54% | 17.54% | — |

### 4.2. Wetland Ecological WRS

The WRS in different scenarios are shown in Table 2. The benefits to the Xianghai wetland and the TID shown in Table 3, were different in different scenarios. As can be seen, the WRS in Scenario IV resulted in the highest benefit. From the perspective of a river basin, the total benefit in Scenario IV was $104.02M which was higher than that of Scenario I (irrigation priority, $99.15M) and Scenario II (wetland priority, $89.46M). The ratio of the benefit to the Xianghai wetland and to the TID in Scenario IV was close to 1, indicating a balance was achieved between the interests of the Xianghai wetland and the TID. That means the win-win solution with consideration of all users of the scarce water resources will result in the best wetland ecological WRS.

**Table 3.** Benefits of the Xianghai wetland and the Taoerhe irrigation district (million$).

| | Wetland Benefit | Irrigation Area Benefit | | | | Total Benefit |
|---|---|---|---|---|---|---|
| | | Crop Output | Crop Quality | Water Cost | Benefit | |
| *Scenario I* | 32.82 | 39.84 | 24.15 | 2.35 | 66.33 | 99.15 |
| *Scenario II* | 89.46 | 0 | 0 | 0 | 0 | 89.46 |
| *Scenario III* | 52.07 | 21.21 | 0.00 | 1.55 | 22.76 | 74.83 |
| *Scenario IV* | 52.07 | 36.35 | 14.05 | 1.55 | 51.95 | 104.02 |

In Scenario I, the priority was to meet WD for irrigation. As shown in Table 2, the available water in the river was 6.928 × 10$^{-2}$ km$^3$ in June which was much smaller than the irrigation water required by the TID (0.156 km$^3$). As a result, the water diverted to the Xianghai wetland was zero in June. The water diverted from the Taoerhe River to the TID varied from May to September with the highest in June and the lowest in September (8.64 × 10$^{-3}$ km$^3$). The total water diverted to the TID

was 0.22 km$^3$ and the remaining 0.127 km$^3$ in the Taoerhe River was then introduced to the Xianghai wetland. The WRS for the Xianghai wetland from May to September was 3.528 × 10$^{-2}$ km$^3$, 0.00 km$^3$, 5.429 × 10$^{-2}$ km$^3$, 2.349 × 10$^{-2}$ km$^3$ and 1.416 × 10$^{-2}$ km$^3$, respectively. In this scenario, since TID has the priority of diverting the water from the Taoerhe River, the benefit to the TID was the highest ($66.33M) among all four scenarios. However, the benefit to the Xianghai wetland is only about half of that to the TID, which means the WRS in this scenario fails to achieve a balance of benefits between the TID and the Xianghai wetland.

In Scenario II, the wetland had the priority of diverting river water. The supply capacity of the Taoerhe River was 0.347 km$^3$ which was less than the Max-WD (0.486 km$^3$ as mentioned in Section 4.1) of the wetland. As a result, all available water would be introduced to the Xianghai wetland with no water left for the TID. The amount diverted to the Xianghai wetland was more than B-WD (0.202 km$^3$), leading to the benefits of $89.46M for the Xianghai wetland. The negative impact to the TID; however, was high with the loss of the benefits of $66.33M due to the use of the groundwater.

In Scenario III, the total B-WD was met to protect wetland ecosystem and the rest of the river water was introduced to the TID without consideration of seasonal variation. As stated in Section 4.1, 0.202 km$^3$ of water should be introduced to the wetland from the Taoerhe River. As the supply capacity of the Taoerhe River in September was only 0.228 km$^3$ which was less than the required amount to meet the monthly B-WD, as a result all water would be diverted to the wetland in September. The rest of the required B-WD (0.179 km$^3$) would be introduced to the Xianghai wetland evenly as 4.476 × 10$^{-2}$ km$^3$ per month from May to August. This WRS cannot ensure the Min-WD was met in each month; for example, the diverted water to the wetland was less than the Min-WD (4.971 × 10$^{-2}$ km$^3$) in May. In this scenario, the total amount diverted to the TID is 0.145 km$^3$ and the surface water DRs in the irrigation area were 27.49%, 28.78%, 0.00%, 39.72% and 100.00% from May to September, respectively. In this case, the total benefits to both the Xianghai wetland and the TID are $74.83M which was the lowest among four scenarios.

In Scenario IV, the seasonal impacts on the farmland were taken into account in water redistribution after meeting the total B-WD. From May to September the WRS of the Xianghai wetland was 4.971 × 10$^{-2}$ km$^3$, 3.989 × 10$^{-2}$ km$^3$, 6.369 × 10$^{-2}$ km$^3$, 3.289 × 10$^{-2}$ km$^3$ and 1.568 × 10$^{-2}$ km$^3$, which can meet both the total B-WD and the monthly Min-WD (Table 2). In this scenario, the total amount diverted to the TID is 0.145 km$^3$ with the DRs of the TID from May to September as 41.83%, 25.65%, 17.54%, 17.54% and 17.54%, respectively. In this case, the total benefit was the highest among four scenarios with the similar benefits to the Xianghai wetland ($52.07M) and to the TID ($51.95M).

In summary, Scenario I sacrificed the goal of wetland ecological restoration for the agricultural development. From the perspective of the Xianghai wetland, the WRS in Scenario II was the most preferred one; however, the cost to the TID was too high. This type of replenishment mode will be applied only in urgent situations to avoid the collapse of the wetland ecological system in the developing countries such as China [41]. The WRS in Scenario III had less benefit than that of Scenario IV because it did not consider the impacts of seasonal water shortage on crop yield and quality. Therefore, given the limited RSC, the WRS of Scenario IV was the best one which cannot only meet the B-WD but also achieve the balance of the benefits between the Xianghai wetland and the TID.

*4.3. Seasonal Effect on the Wetland Ecological Water Replenishment Scheme*

Seasonal effect must be considered in designing WRS because WD, RSC and the water demand of the TID all vary with seasons [42–44].

The monthly WD of the Xianghai wetland changes because of seasonal variation of plant and animal growth, mainly from April to September. The trends of Max-WD, B-WD and Min-WD over a year were similar with one peak occurred in May and one peak occurred in August or September (Figure 3a). The WD in May was the highest because it was the growing season for the plants, but the rainy season had not started yet. The WD in July was relatively low due to the rainy season. WDs of other months were relatively small because of no plant growth during the frozen period.

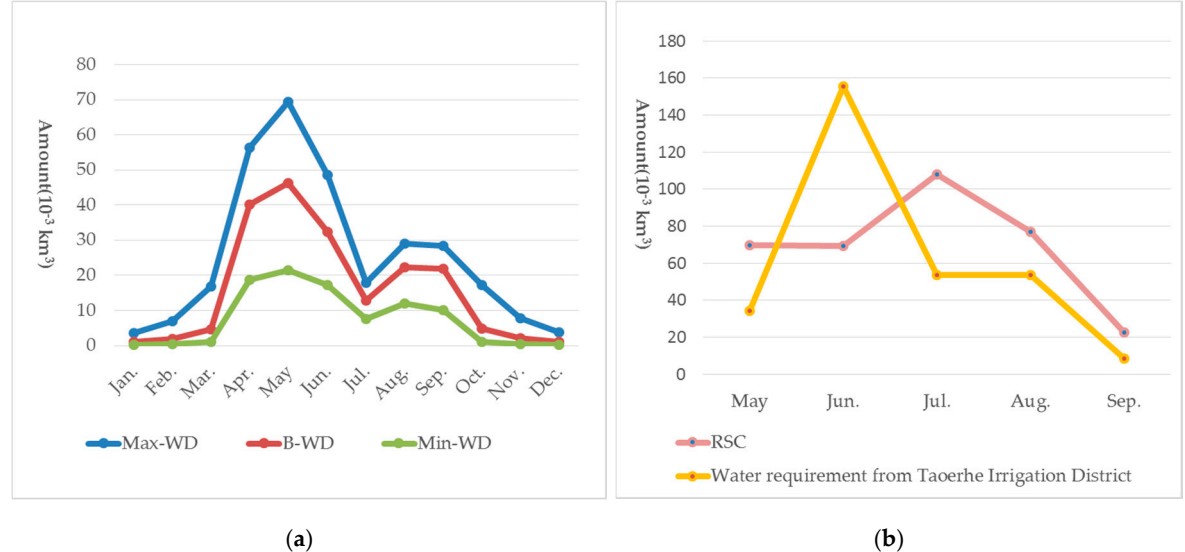

(**a**) (**b**)

**Figure 3.** The seasonal change of the wetland ecological water demand (WD) (Max-WD: maximum water demand; B-WD: baseline water demand; Min-WD: minimum water demand) (**a**), the seasonal change of river water supply capacity (RSC) of the Taoerhe River and the water requirement of the Taoerhe Irrigation district (**b**).

The RSC of the Taoerhe River had seasonal change because of snowmelt and rainfall. The frozen period of the Taoerhe River was from October to April. The snowmelt started in May and the rainy season began in June, leading to the peak capacity (0.108 km$^3$) of the Taoerhe River in July. The rainy season ended in September and the supply capacity of the Taoerhe River consequently reduced to the lowest, $2.28 \times 10^{-2}$ km$^3$.

The WD of the TID also had seasonal change because of crop growth. In addition, the surface water shortage at various growth stages had different effect on crop output and quality (Yu and Ding, 2010). As shown in Figure 3b, the irrigation WD of the TID was the highest in June because the crop was in mid-tilling (MT) stage. From July to August the irrigation WDs were high and relatively constant because of the elongation stage, the efflorescence stage, and the filling stage. The crop reached to the maturation stage in September and the irrigation water requirement was relatively low. Due to the highest WD in MT stage, the greatest impact on crop output occurred in June. Therefore, the WRS with consideration of seasonal effect was able to avoid the negative impact and balance the benefits between the Xianghai wetland and the TID.

From the perspective of a river basin, if the seasonal impact is considered in the WRS design (Scenario IV), the total benefit is the highest, approximately 1.5 times of the total benefits in Scenario III. This is because the WRS in both scenarios (III and IV) meets the total B-WD (0.202 km$^3$), resulting in the same benefit of the Xianghai wetland as $52.07M. The benefit to the TID in Scenario IV; however, is twice of that in Scenario III. Since the crop output reduction rate is much lower in Scenario IV (8.76%) as compared to Scenario III (46.77%), the benefits associated with crop output in scenarios III and IV are $21.21M and $36.35M, respectively. In addition, the impacts on the crop quality are different, indicated by the influence coefficient which is 100% in Scenario III and 41.82% in Scenario IV. Therefore, there is no benefit associated with crop quality in Scenario III and the benefit in Scenario IV is $14.05M.

In summary, the WRS with consideration of seasonal effect not only improves the total benefits of the river basin from $74.83M (Scenario III.) to $104.02M (Scenario IV) but also balances the benefits between the wetland and the irrigation area. It is critical to consider the seasonal effect in the design of WRS.

## 5. Conclusions

This study examined the benefits of four wetland ecological WRSs. The WRS design balancing the benefits between the wetland and the irrigation area achieves the highest total benefit. In addition, the WRS design with consideration of seasonal effects significantly improves the total benefit from $74.83M (Scenario III.) to $104.02M (Scenario IV). The wetland WD, RSC and irrigation requirement of the TID all vary with seasons because of plant growth and the rainy season. The WRS design without considering seasonal effect results in the lowest total benefit and the Min-WD cannot be met in some months leading to the poor wetland ecosystem recovery. The WRS design considering seasonal effects of surface water shortage on crop output and quality results in the relatively high benefit of the TID and the highest total benefit. Therefore, a holistic wetland water replenishment plan should be designed with consideration of all the water users and the seasonal effects. This method can be used to calculate the water use balance among different subjects in the basin, evaluate the ecological service value of wetland ecological water replenishment, and evaluate the loss of crops caused by seasonal water shortage. It can be used in agricultural production water and ecological water contradictory areas.

The concept of balancing the water use among different users in the basin to design a wetland water replenishment plan is general and can be applied to other regions. The method presented in the study can be used for other areas where there is a conflict between wetland ecological WD and irrigation WD to evaluate the wetland ecological service value and the value of crop production by addressing seasonal water shortage via irrigation.

**Author Contributions:** Conceptualization, Q.Z., and H.D.; Data Acquisition, M.X. and Y.C.; Methodology, J.Z. and H.D.; Formal Analysis: X.W. and H.D.; Writing—Original Draft Preparation, H.D.; Writing—Review & Editing, Q.Z.

**Acknowledgments:** This material is based in part upon work supported by the National Science Foundation of China under Grant Numbers51379089.

**Conflicts of Interest:** The authors declare no conflict of interest.

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
