# Peer review of "A Holistic Wetland Ecological Water Replenishment Scheme with Consideration of Seasonal Effect"

_sustainability, doi:10.3390/su11030930_

Reviewer 1 Report

Suggesting some more updated citations here, the ones included here are getting old:

*Lines 30-31-

Thorslund, J., Jarsjo, J., Jaramillo, F., Jawitz, J. W., Manzoni, S., Basu, N. B., et al. (2017). Wetlands as large-scale nature-based solutions: Status and challenges for research, engineering and management. Ecological Engineering, 108, 489–497. https://doi.org/10.1016/j.ecoleng.2017.07.012

*Line 31-33-

Put one or two more exmaples around the world to make this point important. Example: Jaramillo, F., Licero, L., Åhlen, I., Manzoni, S., Rodríguez-Rodríguez, J. A., Guittard, A., et al. (2018). Effects of Hydroclimatic Change and Rehabilitation Activities on Salinity and Mangroves in the Ciénaga Grande de Santa Marta, Colombia. Wetlands, 1–13. https://doi.org/10.1007/s13157-018-1024-7

*Line 60-64-

Campbell, B., Beare, D., Bennett, E., Hall-Spencer, J., Ingram, J., Jaramillo, F., et al. (2017). Agriculture production as a major driver of the Earth system exceeding planetary boundaries. Ecology and Society, 22(4).https://doi.org/10.5751/ES-09595-220408L. 263

Also:

*L. 242 River flows are (Q) are given in rates, not in volumes. Ex. m3/s or mm/yr

*L. 250 What do you mean by "3.4 Date sources"? I do not understand

*L. 336. Million m3, better km3?

**Mention (Table 1), otherwise if feels you are not giving any reference. In Table 1 make a consistent column to put the units.

*Be consistent with the units of area along the manuscript. Million m2, million ha. I would use in all just km2.

*Figure 2 is sloppy. What is amount? No units. Improve quality of graphs.

 Author Response

Response: We appreciate the positive feedback and constructive criticisms for improving the manuscript’s quality prior to publication. This letter contains detailed responses to the reviewer’s comment.

 Point 1: Suggesting some more updated citations here, the ones included here are getting old:

×Lines 30-31- Thorslund, J., Jarsjo, J., Jaramillo, F., Jawitz, J. W., Manzoni, S., Basu, N. B., et al. (2017). Wetlands as large-scale nature-based solutions: Status and challenges for research, engineering and management. Ecological Engineering,108, 489–497. https://doi.org/10.1016/j.ecoleng.2017.07.012

Line 31-33- Put one or two more exmaples around the world to make this point important. Example: Jaramillo, F., Licero, L., Åhlen, I., Manzoni, S., Rodríguez-Rodríguez, J. A., Guittard, A., et al. (2018). Effects of Hydroclimatic Change and Rehabilitation Activities on Salinity and Mangroves in the Ciénaga Grande de Santa Marta, Colombia. Wetlands, 1–13. https://doi.org/10.1007/s13157-018-1024-7

*Line 60-64-Campbell, B., Beare, D., Bennett, E., Hall-Spencer, J., Ingram, J., Jaramillo, F., et al. (2017). Agriculture production as a major driver of the Earth system exceeding planetary boundaries. Ecology and Society,22(4).https://doi.org/10.5751/ES-09595-220408L. 263

Response 1: The authors would like to thank reviewer for suggesting the above updated citations. We have reviewed the recommended papers and added the citations as suggested in the revised manuscript. References 48-50 have been added to the reference list.

Point 2: L. 242 River flows are (Q) are given in rates, not in volumes. Ex.  m3/s  or mm/yr*L.

Response 2: We agree with the reviewer that river flows should be given in rates not in volumes. Q, in this study, is the total amount of water in the river during a period of time (month or season), so the unit is m3. To avoid the confusion, the relevant nomenclature has been revised as “ Qi is the total amount of water in the river in the month or season i (km3); Qeco,i is the amount of water required to maintain environmental flow of the river in the month or season i (km3).”

Point 3:L. 250 What do you mean by "3.4 Date sources"? I do not understand

Response 3:T his is a spelling error. The word “Date” has been replaced by “Data”.

 Point 4: L. 336. Million m3, better km3?

Response 4: The unit of km3 has been used in the revised manuscript.

Point 5:Mention (Table 1), otherwise if feels you are not giving any reference. In Table 1 make a consistent column to put the units.

Response 5: Table 1 was mentioned in Section 3.4 as “The data and parameters required in Equations (1)-(6) are compiled in Table 1”. A new column for units has been added in Table 1 of the revised manuscript

Point 6: Be consistent with the units of area along the manuscript. Million m2, million ha. I would use in all just km2.

Response 6: The unit of area has been revised as km2 throughout the manuscript.

Point 7: Figure 2 is sloppy. What is amount? No units. Improve quality of graphs.

Response 7: The revised manuscript improves figure 3 on line 273 by adding units.

Reviewer 2 Report

This MS “A holistic wetland ecological water replenishment scheme with consideration of seasonal effect” intended to compare total benefits under different water allocation scenarios. This study is interesting and worthy of consideration for publication in this journal.

Major comments:

1.      The calculation processes are not easy to understand. If a flowchart of the calculation flow is added, this can increase readability.

2.      In this MS, the authors used both present and past tenses for stating results. I suggest using the past tense. For example, line 18-19, The results show that …

3.      Discussions of relevant results in literature are expected. Only one citation is present in the result and discussion section. More citations are expected.

4.      The space between words is missing. For example, line 35, Sciences,2011)

5.      The format of references is inconsistent. For example, line 406, Alvaro Calzadilla, …

Specific comments:

1.      Line 19: … showed …

2.      Line 87: … at sand plants?

3.      Line 126: (Liu et al. 2008) is not in the references.

4.      Line 218: It may need descriptions of average snow and rain annually and their periods.

5.      Line 245: The line spacing is different from previous paragraphs.

6.      Table 1: Terzaghi Dam of Canada et al.?

7.      Line 276: … natural Inflow …

8.      Figure 2: move it close to its text?

Author Response

Response: We appreciate the completed review work and hope that these revisions will address the reviewer’s comments and suggestions. The revised manuscript is now improved and this letter contains detailed responses to the reviewer’s comment.

Comments and Suggestions for Authors

This MS “A holistic wetland ecological water replenishment scheme with consideration of seasonal effect” intended to compare total benefits under different water allocation scenarios. This study is interesting and worthy of consideration for publication in this journal.

Response: The authors would like to thank the reviewer for acknowledging the contribution of this study.

Major comments:

Point 1: The calculation processes are not easy to understand. If a flowchart of the calculation flow is added, this can increase readability.

Response 1: We appreciate your suggestion and have added the following flowchart to the manuscript to make the calculation easier to understand. It reads” The method system presented in the paper was described as Fig.1. According to the calculation of wetland ecological water demand, the replenishment of wetland water from the water inflow was proposed. Meanwhile, the river water supply capacity in different seasons was simulated. The surplus water supply capacity was defined as the difference between the water supply capacity and the water consumption. In the case of the surplus water supply capacity more than replenishment of wetland, the river water was directly transmitted into wetland. Otherwise, the river water used in irrigation was converted to replenish the wetland. The ground water was used in irrigation instead of river water. Therefore, the water transmission led to the benefits and losses for wetland and farmland accordingly. Based on the benefits and losses, the river water was reallocated.”

Point 2: In this MS, the authors used both present and past tenses for stating results. I suggest using the past tense. For example, line 18-19. The results show that …

Response 2: The tense for stating results has been changed to the past tense in lines 18-19 and section 4.

Point 3: Discussions of relevant results in literature are expected. Only one citation is present in the result and discussion section. More citations are expected.

Response 3: We have added the citations in line 341 and 350 which can support the relevant results.

Point 4:The space between words is missing. For example, line 35, Sciences,2011)

Response 4: Spaces between words have been added. This may be a Microsoft Word version problem.

Point 5: The format of references is inconsistent. For example, line 406, Alvaro Calzadilla,...

Response 5: All references have been revised to keep the format consistent.

Point 6: Specific comments:

1.Line 19: … showed …

2. Line 87: … at sand plants?

3.Line 126: (Liu et al. 2008) is not in the references.

4. Line 218: It may need descriptions of average snow and rain annually and their periods.

5.Line 245: The line spacing is different from previous paragraphs.

6.Table 1: Terzaghi Dam of Canada et al.?

7.Line 276: … natural Inflow …

8.Figure 2: move it close to its text?

Response 6:

1. The word “show” has been replaced byshowed”in the manuscript.

2. The word “habitat sand plants” has been replaced by“habitats and plants”in the manuscript.

3.Reference 47 was added to the manuscript.Liu, X.H.; Lv, X.G.; Jiang, M.; Shang, L.N.; Wang, X.G. 2008. Research on the valuation of wetland ecosystem services. Acta Ecologica Sinica 28(11),5625-5631 (in Chinese, with English abstract).”

4.The average precipitation is in line 223 “the annual average precipitation is 360.6 mm” .The rainy season is from June to August. With regard to snowfall, our study period is from June to October and has been covered by natural incoming water.

5.The line spacing has been correct and consistent with previous paragraphs.

6.The correct citation was added “Wang (2012)”.

7. The word “natural Inflow” has been replaced by“natural inflow”in the manuscript.

8.Figure 2 is already next to its corresponding text.
